# SHE: A Fast and Accurate Deep Neural Network for Encrypted Data*

**Qian Lou**
Indiana University Bloomington
louqian@iu.edu

Lei Jiang
Indiana University Bloomington
jiang60@iu.edu

## Abstract

Homomorphic Encryption (HE) is one of the most promising security solutions to emerging Machine Learning as a Service (MLaaS). Leveled-HE (LHE)-enabled Convolutional Neural Networks (LHECNNs) are proposed to implement MLaaS to avoid large bootstrapping overhead. However, prior LHECNNs have to pay significant computing overhead but achieve only low inference accuracy, due to their polynomial approximation activations and poolings. Stacking many polynomial approximation activation layers in a network greatly reduces inference accuracy, since the polynomial approximation activation errors lead to a low distortion of the output distribution of the next batch normalization layer. So the polynomial approximation activations and poolings have become the obstacle to a fast and accurate LHECNN model.

In this paper, we propose a **S**hift-accumulation-based **LHE**-enabled deep neural network (SHE) for fast and accurate inferences on encrypted data. We use the binary-operation-friendly Leveled Fast Homomorphic Encryption over Torus (LTFHE) encryption scheme to implement $ReLU$ activations and max poolings. We also adopt the logarithmic quantization to accelerate inferences by replacing expensive LTFHE multiplications with cheap LTFHE shifts. We propose a mixed bitwidth accumulator to accelerate accumulations. Since the LTFHE $ReLU$ activations, max poolings, shifts and accumulations have small multiplicative depth overhead, SHE can implement much deeper network architectures with more convolutional and activation layers. Our experimental results show SHE achieves the state-of-the-art inference accuracy and reduces the inference latency by $76.21\% \sim 94.23\%$ over prior LHECNNs on MNIST and CIFAR-10. The source code of SHE is available at https://github.com/qianlou/SHE.

## 1 Introduction

Machine Learning as a Service (MLaaS) is an effective way for clients to run their computationally expensive Convolutional Neural Network (CNN) inferences [1] on powerful cloud servers. During MLaaS, cloud servers can access clients' raw data, and hence potentially introduce privacy risks. So there is a strong and urgent need to ensure the confidentiality of healthcare records, financial data and other sensitive information of clients uploaded to cloud servers. Recent works [1, 2, 3, 4, 5] employ Leveled Homomorphic Encryption (LHE) to do CNN inferences over encrypted data. During the LHE-enabled MLaaS, a client encrypts the sensitive data and sends the encrypted data to a server. Since only the client has the private key, the server cannot decrypt the input nor the output. The server produces an encrypted inference output and returns it to the client. The client privacy is preserved in this pipeline for both inputs and outputs.

However, prior LHE-enabled CNNs (LHECNNs) [1, 2, 3, 4, 5] suffer from low inference accuracy and long inference latency. TAPAS [5] and DiNN [1] adopt only 1-bit weights, inputs and $sign$

activations. They degrade $3\% \sim 6.2\%$ inference accuracy even on tiny hand-written digit dataset MNIST. Since HE supports only polynomial computations, CryptoNet [2], NED [3], n-GraphHE [6], Lola [7] and Faster Cryptonet [4] have to use the low-degree polynomial activations rather than $ReLU$, and thus fail to obtain the state-of-the-art inference accuracy. For instance, Faster Cryptonet achieves only 76.72% inference accuracy on CIFAR-10 dataset, while the inference accuracy of an unencrypted CNN model with $ReLU$ activations is 93.72%. Although it is possible to improve the encrypted inference accuracy by enlarging the degree of polynomial approximation activations, the computing overhead increases exponentially with the degree. With even a degree-2 $square$ activation, prior LHECNNs spend hundreds of seconds in doing an inference on an encrypted MNIST image.

Moreover, the polynomial approximation activation (PAA) is not compatible with a deep network consisting of many convolutional and activation layers. Stacking many convolutional and PAA layers in a CNN actually decreases inference accuracy [8], since the PAA approximation errors lead to a low distortion of the output distribution of the next batch normalization layer. As a result, no prior LHECNN fully supports deep CNNs that can infer the large ImageNet. For instance, Faster Cryptonet [4] can compute only a part (i.e., 3 convolutional layers with $square$ activations) of a CNN inferring ImageNet on a server but leave the other part of the CNN to the client.

## 2 Background and Motivation

**Threat model**. In the MLaaS paradigm, a risk inherent to data transmission exists when clients send sensitive input data or servers return sensitive output results back to clients. Although a strongly encryption scheme can be used to encrypt data sent to cloud, untrusted servers [2, 3, 4] can make data leakage happen. HE is one of the most promising encryption techniques to enable a server to do CNN inference over encrypted data. A client sends encrypted data to a server performing encrypted inference without decrypting the data or accessing the client's private key. Only the client can decrypt the inference result using the private key.

**Homomorphic Encryption**. An encryption scheme defines an encryption function $\epsilon()$ encoding data (plaintexts) to ciphertexts (encrypted data), and a decryption function $\delta()$ mapping ciphertexts back to plaintexts (original data). In a public-key encryption, the ciphertexts $x$ can be computed as $\epsilon(x, k_{pub})$, where $k_{pub}$ is the public key. The decryption can be done through $\delta(\epsilon(x, k_{pub}), k_{pri}) = x$, where $k_{pri}$ is the private key. An encryption scheme is *homomorphic* in an operation $\odot$ if there is another operation $\oplus$ such that $\epsilon(x, k_{pub}) \oplus \epsilon(y, k_{pub}) = \epsilon(x \odot y, k_{pub})$. To prevent threats on untrusted servers, fully HE [9] enables an unlimited number of computations over ciphertexts. However, each computation introduces noise into the ciphertexts. After a certain number of computations, the noise grows so large that the ciphertexts cannot be decrypted successfully. A *bootstrapping* [9] is required to keep the noise in check without decrypting. Unfortunately, the bootstrapping is extremely slow, due to its high computational complexity. Leveled HE (LHE) [2] is proposed to accelerate encrypted computations without bootstrapping. But LHE can compute polynomial functions of only a maximal degree on the ciphertexts. Before applying LHE, the complexity of the target arithmetic circuit processing the ciphertexts must be known in advance. Compared to Multiple Party Computation (MPC) [10, 11], LHE has much less communication overhead. For example, a MPC-based MLaaS DeepSecure [11] has to exchange 722GB data between the client and the server for only a 5-layer CNN inference on one MNIST image.

**TFHE**. TFHE [10] is a HE cryptosystem that expresses ciphertexts over the torus modulo 1. It supports both fully and leveled HE schemes. Like the other HE cryptosystems including BFV/BGV [7], FV-RNS [4] and HEAAN [6], TFHE is also based on the Ring-LWE, but it can perform very fast binary operations over encrypted binary bits. Therefore, unlike the other HE cryptosystems approximating the activation by expensive polynomial operations, TFHE can naturally implement $ReLU$ activations and max poolings by Boolean operations. In this paper, we use TFHE for SHE without the batching technique [10]. Although the ciphertext batching may greatly improve the LHECNN inference throughput by packing multiple (e.g. 8K) datasets into a homomorphic operation, the batching has to select more numerous and restricted NTT points, force specific computations away from NTT, and add large computing overhead [4]. Moreover, it is difficult for a client to batch 8K requests together sharing the same secret key. In fact, a client often needs to do inferences on only few images [7].

**Motivation**. As Figure 1 describes, prior LHECNNs suffer from low inference accuracy and long inference latency, because of the polynomial approximation activations (PAAs) and poolings. CryptoNet (CNT) [2] and Faster CryptoNet (FCN) [4] add a batch normalization layer before each

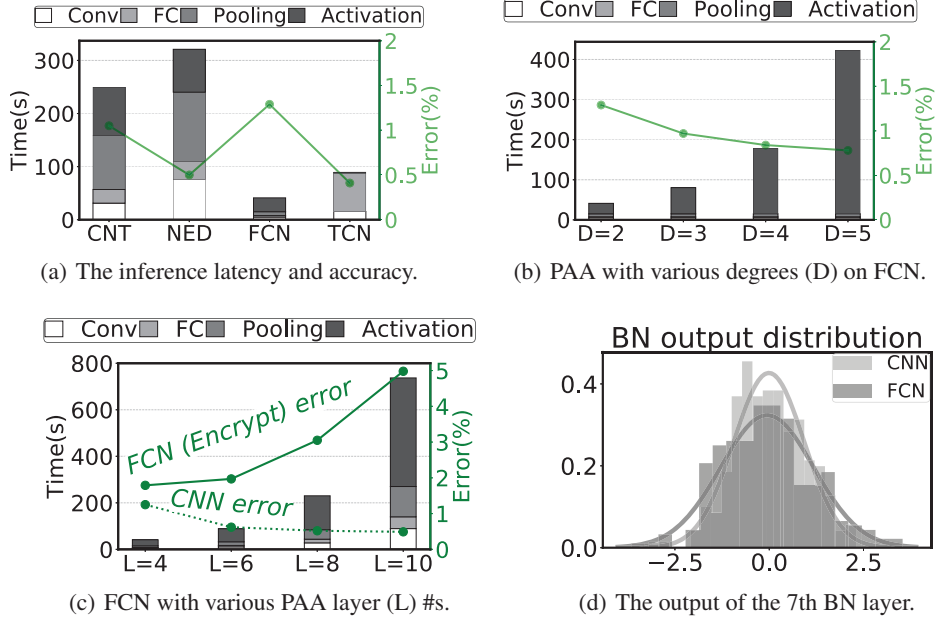

(a) The inference latency and accuracy.

(b) PAA with various degrees (D) on FCN.

(c) FCN with various PAA layer (L) #s.

(d) The output of the 7th BN layer.

Figure 1: The inference latency and accuracy of prior LHECNNs inferring MNIST (Conv: convolutional layers; FC: fully-connected layers; Pooling: pooling layers; and Activation: batch normalization (BN) and activation layers).

| Name | Cryptosys. | $ReLU$ | MaxPool | NoConv | DeepNet |
|---|---|---|---|---|---|
| CNT[2] | YASHE | ✗ | ✗ | ✗ | ✗ |
| NED[3] | BGV | ✗ | ✗ | ✗ | ✗ |
| FCN[4] | FV-RNS | ✗ | ✗ | ✗ | ✗ |
| DiNN[1] | TFHE | ✗ | ✗ | ✓ | ✗ |
| TAPAS[5] | TFHE | ✗ | ✗ | ✓ | ✗ |
| GHE[6] | HEAAN | ✗ | ✗ | ✗ | ✗ |
| Lola[7] | BFV | ✗ | ✗ | ✗ | ✗ |
| SHE | TFHE | ✓ | ✓ | ✓ | ✓ |

Table 1: The comparison of LHECNNs.

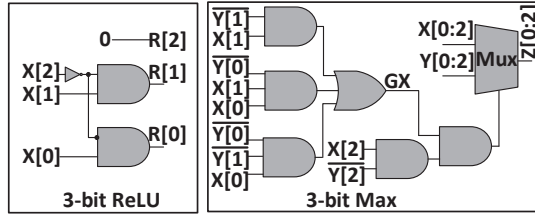

Figure 2: A 3-bit ReLU unit ($X$: input; $R$: output) and a 3-bit max pooling unit ($X$ and $Y$: inputs; $Z$: output).

activation layer. They use $square$ (polynomials with degree-2) activations replace $ReLU$s, and led mean poolings to replace max poolings. As Figure 1(a) shows, the $square$ activation reduces inference accuracy by $0.82\%$ compared to unencrypted models on MNIST. Although increasing the degree of PAAs improves their inference accuracy, the computing overhead of PAA layers exponentially increases as shown in Figure 1(b). NED [3] uses high-degree PAAs to obtain nearly no accuracy loss, but its inference latency is much longer than CNT. For CNT, NED, and FCN, PAA layers occupy $65.7\% \sim 81.2\%$ of the total inference latency. Integrating more convolutional, and PAA layers in a LHECNN cannot improve inference accuracy neither. As Figure 1(c) shows, an unencrypted CNN achieves higher inference accuracy with more layers, but the inference accuracy of the LHE-enabled FCN decreases when having more convolutional and PAA layers. This is because the $square$ approximation errors result in a low distortion of the output distribution of the next batch normalization layer. Figure 1(d) illustrates such an output distribution distortion of the 7th batch normalization layer in a 10-layer LHE-enabled FCN.

**Comparison against prior works**. Table 1 compares our SHE against prior LHECNN models including CryptoNet (CNT) [2], NED [3], n-GraphHE (GHE) [6], Lola [7] and Faster CryptoNet (FCN) [4], DiNN [1] and TAPAS [5]. Due to the limitation of non-TFHE schemes, CNT, NED, GHE, Lola and FCN cannot support $ReLU$ and max pooling operations. They are also slowed down by computationally expensive homomorphic multiplications. Particularly, Lola encodes each weight of a LHECNN as a separate message for encryption batches to reduce inference latency, but its inference latency is still significant due to the PAA computations and homomorphic multiplications. Although DiNN [1] and TAPAS [5] binarize weights, inputs and activations to avoid homomorphic multiplications, their inference accuracy decreases significantly.

## 3 SHE

### 3.1 ReLU Activation and Max Pooling

The rectifier, $ReLU(x) = max(x, 0)$, is one of the most well-known activation functions, while the max pooling is a sample-based discretization process heavily used in state-of-the-art CNN models. Prior BF/V [2] and FV-RNS [4]-based LHECNNs approximate both $ReLU(x)$ and max pooling by linear polynomials leading to significant inference accuracy degradation and huge computing overhead. Prior TFHE-based CNN models [1, 5] use the 1-bit $sign()$ function to implement binarized activations that introduce large inference accuracy loss.

In this paper, we propose an accurate homomorphic $ReLU$ function and a homomorphic max pooling operation. TFHE can implement any 2-input binary homomorphic operation [9, 10], e.g., AND, OR, NOT, and MUX, over encrypted binary data by a deterministic automaton, a Boolean gate or a look-up table. In this way, as Figure 2 exhibits, we can connect the TFHE homomorphic Boolean gates to construct a $ReLU$ unit and a max pooling unit. A > 2-input TFHE gate can be divided into multiple 2-input TFHE gates.

### 3.2 Logarithmic Quantization

When we implemented Faster CryptoNet (FCN) through TFHE Boolean gates, as Figure 1(a) shows, we observe that the TFHE-version FCN model (TCN) although has the same network topology as FCN, its inference latency is much larger. This is because TFHE is not designed and optimized for homomorphic matrix multiplications or other repetitive tasks. So the convolutional and fully-connected layers of TCN have become the new latency bottleneck.

To reduce the computing overhead of homomorphic matrix multiplications, we logarithmically quantize the floating-point weights into their power-of-2 representations [12], so that we can replace all homomorphic multiplications in a LHECNN inference by homomorphic shifts and homomorphic accumulations. Prior works [13, 12] suggest the logarithmic quantization even with weight pruning still achieves the same inference accuracy as full-precision models. In a logarithmically quantized CNN model, $weight^T * input$ is approximately equivalent to $\sum_{i=1}^{n} input_i \times 2^{wegihtQi}$, and hence can be converted to $\sum_{i=1}^{n} binaryshift(input_i, weightQ_i)$, where $weightQ_i = Quantize(log_2(weight_i))$, $Quantize(x)$ quantizes $x$ to the closest integer and $binaryshift(a, b)$ shifts $a$ by $b$ bits in fixed-point arithmetic.

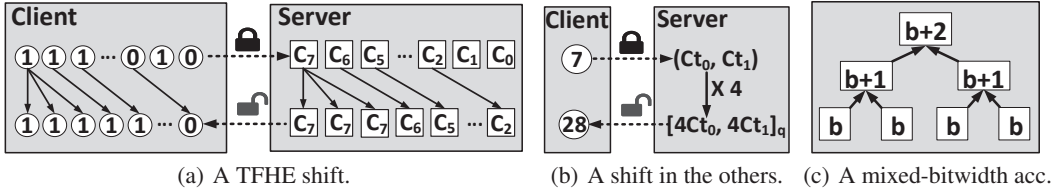

(a) A TFHE shift.   (b) A shift in the others.   (c) A mixed-bitwidth acc.

Figure 3: The logarithmic quantization in different HE cryptosystems.

As Figure 3(a) highlights, a TFHE arithmetic shift operation is computationally cheap. TFHE encrypts the plaintext bit by bit. Moreover, it keeps the order of the ciphertext the same as that of the plaintext. A TFHE arithmetic shift just copies the encrypted sign bit and the encrypted data to the shifted positions. It takes only $\sim 100ns$ for a CPU core to complete a TFHE arithmetic shift operation. On the contrary, in the other HE schemes, e.g., BFV/BGV [7], FV-RNS [4] and HEAAN [6], a homomorphic arithmetic shift is equivalent to a homomorphic multiplication, as they encrypt each floating point number of plaintext as a whole shown in Figure 3(b). Compared to TFHE, a homomorphic arithmetic shift in the other cryptosystems are much more computationally expensive.

### 3.3 Mixed Bitwidth Accumulator

Besides homomorphic shift operations, the logarithmically quantized CNN models also require accumulations to do inferences. The computing overhead of a TFHE adder is proportional to its bitwidth. For instance, compared to its 4-bit counterpart, an 8-bit TFHE adder doubles the computing overhead. We quantize the weights into their 5-bit power-of-2 representations, since recent works [12] show a 5-bit logarithmically quantized AlexNet model degrades the inference accuracy by only $< 0.6\%$. However, the accumulation intermediate results have to be represented by 16 bits. Otherwise, the inference accuracy may dramatically decrease. Accumulating 5-bit shifted results through 16-bit TFHE adders is computationally expensive.

|  | Topology | MD overhead (K) | Total | Acc(%) |
|---|---|---|---|---|
| FCN | C-B-A-P-C-P-F-B-A-F | 1.4-0.3-3-2.4-1.4-2.4-3.1-0.3-4.6-2 | 21K | 98.71 |
| TCN | C-B-A-P-C-P-F-B-A-F | 1.4-0.3-0.1-0.2-0.2-0.2-3.1-0.3-0.1-2 | 9.0K | 99.54 |
| SHE | C-B-A-P-C-P-F-B-A-F | 0.2-0.3-0.1-0.2-0.2-0.2-0.4-0.3-0.1-0.3 | 2.0K | 99.54 |
| DSHE | [C-B-A-C-B-A-P]×4-F-F | [0.2-0.3-0.1-0.2-0.3-0.1-0.1]x4-0.7-0.5 | 6.8K | 99.77 |

Table 2: The multiplicative depth (MD) overhead of prior LHECNNs. In the column of topology, $C$ means a convolution layer; $B$ is a batch normalization layer; $A$ indicates an activation layer; $P$ denotes a pooling layer; and $F$ is a fully-connected layer. Acc is the inference accuracy.

Therefore, we propose a mixed bitwidth accumulator shown in Figure 3(c) to avoid the unnecessary computing overhead. A mixed bitwidth accumulator is an adder tree, where each node is a TFHE adder. And TFHE adders at different levels of the tree have distinctive bitwidths. At the bottom level ($layer_0$) of the tree, we use $b$-bit TFHE adders, where $b$ is 5. Instead of 16-bits adders, $layer_1$ of the tree has ($b + 1$)-bit TFHE adders, since the sum of two 5-bit integers can have at most 6 bits. The $n_{th}$ level of the mixed bitwidth accumulator consists of ($b + n$)-bit TFHE adders.

### 3.4 Deeper Neural Network

A LHE scheme with a set of predefined parameters allows only a fixed *multiplicative depth* (MD) budget, which is the maximal number of consecutively executed homomorphic AND gates in the LHE Boolean circuit [14]. The total MD overhead of an $n$-layer LHECNN is the sum of the MD overhead of each layer, i.e., $\sum_{i=0}^{n} LMD_i$, where $LMD_i$ is the MD overhead of the $i_{th}$ layer. $LMD_i$ can be defined as

$$IN \cdot log_2(KC^2) \cdot (DA[a] + DM[b]) + DR[c] + log_2(KP^2) \cdot DM[d] \tag{1}$$

where $IN$ is the input channel #; $KC$ means the weight kernel size; $DA[a]$ is the MD overhead of an $a$-bit adder; $DM[b]$ indicates the MD overhead of a $b$-bit multiplier; $DR[c]$ is the MD overhead of a $c$-bit $ReLU$ unit; $KP$ denotes the pooling kernel size; $DM[d]$ is the MD overhead of a $d$-bit max pooling unit. The LHECNN model must guarantee its MD budget no smaller than its total MD overhead, otherwise, its inference output cannot be successfully decrypted. Although it is possible to enlarge the LHECNN MD budget by re-designing LHE parameters, the new LHE parameters increase the ciphertext size and prolong the overall inference latency of the LHECNN. Considering the inference latency of prior LHECNNs has already been hundreds of seconds, enlarging the LHECNN MD budget is not an attractive way to achieve a deeper network architecture. In fact, prior LHECNNs have huge MD overhead in each layer, since they perform matrix multiplications and *square* activations, i.e., $DM[b]$ and $DR[c]$ are large. As Table 2 shows, in a FCN inferring MNIST, each convolutional or fully-connected layer has $1K \sim 2K$ MD overhead. In contrast, the activation or pool layer has $2 \sim 5K$ MD overhead. As a result, FCN can support only shallow architectures with less layers. Moreover, FCN has to reduce the number of activation layers by using larger weight kernel sizes and adding more fully-connected layers. As a result, FCN achieves only 98.71% accuracy when inferring MNIST.

To obtain a deep network architecture with more layers under a fixed MD budget, we need to reduce the MD overhead of each layer ($LMD_i$). We created a network with the same architecture as FCN (TCN) by the LTFHE cryptosystem, i.e., we used our proposed $ReLU$ units, max pooling units, TFHE adders and multipliers to implement the network architecture of FCN in TCN. The total MD overhead of TCN inferring on MNIST is reduced to $9.0K$ in Table 2, since the MD overhead of each activation or pooling layer is greatly reduced by TFHE. When we further applied the logarithmic quantization, TFHE shifters and mixed bitwidth accumulators, the total MD overhead of SHE decreases to only 2032. This is because the TFHE shifter has 0 MD overhead, while our mixed bitwidth accumulator also has small MD overhead along the carry path. We further propose a deeper SHE (DSHE) architecture by adding more convolutional and activation layers. Compared to the 10-layer FCN, our 30-layer DSHE increases the MNIST inference accuracy to 99.77% with a MD overhead of $6.8K$.

## 4 Experimental Methodology

**TFHE setting and security analysis**. Our techniques are implemented by TFHE [9]. We used all 3 levels of TFHE in the LHE mode. The level-0 TLWE has the minimal noise standard variation $\underline{\alpha} = 6.10 \cdot 10^{-5}$, the count of coefficients $\underline{n} = 500$, and the security degree $\underline{\lambda} = 194$ [9]. The level-1 TRLWE configures the minimal noise standard variation $\alpha = 3.29 \cdot 10^{-10}$, the count of coefficients $n = 1024$, and the security degree $\lambda = 152$. The level-3 TRGSW sets the minimal

noise standard variation $\overline{\alpha} = 1.42 \cdot 10^{-10}$, the count of coefficients $\overline{n} = 2048$, the security degree $\overline{\lambda} = 289$. We adopted the same key-switching parameters as [9]. Therefore, SHE allows 32K depth of homomorphic AND primitive in the LHE mode [9]. The security degree of SHE is $\lambda = 152$.

**Simulation, benchmark and dataset**. We ran all experiments on an Intel Xeon E7-4850 CPU with 1TB DRAM. Our datasets include MNIST, CIFAR-10, ImageNet and the diabetic retinopathy dataset [4] (denoted by medicare). Medicare comprises retinal fundus images labeled by the condition severity including 'none', 'mild', 'moderate', 'severe', or 'proliferative'. We scaled the image of medicare to the same size of ImageNet, 224×224. We also adopted the Penn Treebank dataset [15] to evaluate a LTFHE-enabled LSTM model.

| Dataset | Network Topology |
|---|---|
| MNIST | SHE: C-B-A-P-C-P-F-B-A-F [2]; DSHE: [C-B-A-C-B-A-P]×4-F-F [8] |
| CIFAR-10 | SHE:[C-B-A-C-B-A-P]×3-F-F [8]; DSHE: ResNet-18 |
| Penn Treebank | LSTM: time-step 25, 1 300-unit layer; $ReLU$; [15] |
| ImageNet & Medical | AlexNet, ResNet-18 and ShuffleNet [16] |

Table 3: Network topology ($C$ means a convolution layer; $A$ is an activation layer; $B$ is a batch normalization layer; $P$ indicates a pooling layer; and $F$ is a fully-connected layer).

**Network architecture**. The network architectures we estimated for various datasets are summarized in Table 3. For MNIST, we set SHE the same as CNT [2] and DSHE the same as [8]. For CIFAR-10, we adopted the architecture of SHE from [8] and ResNet-18 as DSHE. To evaluate Penn Treebank, we used the LSTM architecture from [15], where the activations of all LSTM gates are converted to $ReLU$. Compared to the original LSTM with different activation functions, e.g., $ReLU$, $sigmoid$ and $tanh$, the LSTM with all $ReLU$ has only $< 1\%$ accuracy degradation [15]. For ImageNet and Medical datasets, we adopted AlexNet, ResNet-18 and ShuffleNet [16]. For all models, we quantized weights into 5-bit power-of-2 representations, and converted inputs and activations to 5-bits fixed point numbers with little accuracy loss [12].

## 5 Results and Analysis

We report the inference latency and accuracy of LHECNN models, the numbers of homomorphic operations (HOPs), and homomorphic gate operations (HGOPs) in Table 4~ 7. Each homomorphic operation can be divided into multiple homomorphic gate operations in the TFHE cryptosystem. HOPs and HGOPs are independent of the hardware setting and software parallelization. Specifically, the HOPs include additions, multiplications, shifts and comparisons between ciphertext and plaintext. The multiplication between two ciphertexts ($CC_{mult}$) is the most computationally expensive operation among all HOPs, since it requires the largest number of homomorphic gate operations (HGOPs).

We compared SHE against eight the state-of-the-art LHECNN schemes including CryptoNets (CNT) [2], NED [3], Faster CryptoNets (FCN) [4], DiNN [1], nGraph-HE1 (GHE) [6] and Lola [7]. More details can be viewed in Table 1.

| Scheme | HOPs | $CC_{Add}$ | $PC_{Mult}$ | $CC_{Mult}$ | $PC_{Shift}$ | $CC_{Com}$ | HGOPs | Depth | MSize | EDL(s) | FIL(s) | LIL(s) | Acc(%) |
|---|---|---|---|---|---|---|---|---|---|---|---|---|---|
| CNT | 612K | 312K | 296K | 945 | 0 | 0 | - | 134K | 368MB | 47.5 | - | 250 | 98.95 |
| FCN | 67K | 38K | 24K | 945 | 0 | 0 | - | 21K | 411MB | 6.7 | - | 39.1 | 98.71 |
| GHE | 612K | 312K | 296K | 945 | 0 | 0 | - | - | - | 47.5 | - | 135 | 98.95 |
| Lola | 24.6K | 12.5K | 10.5K | 1.6K | 0 | 0 | - | - | - | 4.4 | - | 2.2 | 98.95 |
| DiNN | 16K | 8K | 8K | 40 | 0 | 0 | 40K | 80 | 66MB | 0.0002 | - | 0.49 | 93.71 |
| NED | 4.7M | 2.3M | 2.3M | 1.6K | 0 | 0 | - | 172K | 337MB | 16.7 | - | 320 | 99.52 |
| TCN | 42K | 19K | 19K | 0 | 0 | 3K | 8.3M | 8.8K | 123MB | 0.0014 | 108K† | 88.6 | 99.54 |
| **SHE** | 23K | 19K | 945 | 0 | 19K | 3K | 856K | 2.0K | 123MB | 0.0014 | 11K† | **9.3** | **99.54** |
| **DSHE** | 613K | 304K | 5K | 0 | 304K | 5K | 11.6M | 6.2K | 123MB | 0.0014 | 149K† | 124.9 | **99.77** |

Table 4: The MNIST results (HOPs: homomorphic operations; HGOPs: homomorphic gate operations; Depth: multiplicative gate depth; MSize: ciphertext message size; EDL: encryption and decryption latency; Acc: inference accuracy; FIL: fully TFHE inference latency; LIL: leveled TFHE inference latency; $CC_{Add}$: # of additions between two ciphertexts; $PC_{Mult}$: # of multiplications between a plaintext and a ciphertext; $CC_{Mult}$: # of multiplications between two ciphertexts; $PC_{Shift}$: # of shifts between a plaintext and a ciphertext; $CC_{Com}$: # of comparisons (including $ReLU$ and max pooling) between two ciphertexts. † denotes that we ran only its first 3 layers and made FIL values by projections).

## 5.1 MNIST

As Table 4 shows, CNT obtains $98.95\%$ accuracy by degree-2 polynomial approximation activations (PAAs). The PAA errors prevent CNT from approaching the unencrypted state-of-the-art inference accuracy on MNIST. FCN slightly degrades the inference accuracy by $0.24\%$ but shortens the inference latency by $84.3\%$ by quantizing CNT and pruning its redundant weights. The weight pruning of FCN significantly decreases the numbers of $CC_{Add}$ and $PC_{Mult}$. In contrast, FCN keeps the same number of activations, so it has the same number of $CC_{Mult}$ occurring during only activations. Both GHE and Lola reduce the inference latency by batching optimizations, but they still have to use PAAs and achieve the same inference accuracy as CNT. DiNN uses the TFHE cryptosystem and quantizes all network parameters (i.e., weights, inputs, and activations) to 1-bit, so it reduces the inference accuracy by $5.29\%$. Through increasing the degree of polynomial approximation activations, NED improves the inference accuracy by $0.58\%$ over CNT. However, compared to CNT, it prolongs the inference latency by $28.2\%$, because it has much more $CC_{Add}$, $PC_{Mult}$ and $CC_{Mult}$ to compute during an inference.

We used the TFHE cryptosystem to implement the network architecture of FCN (denoted by TCN) by using LTFHE-based $ReLU$ activations, max poolings and matrix multiplications. Due to the $ReLU$ activations and max poolings, TCN enhances the inference accuracy by $0.6\%$ over FCN. But compared to FCN, it slows down the inference by $126.6\%$, since TFHE is not suitable to implement massive repetitive matrix multiplications. To create the SHE scheme, we further use power-of-2 quantized weights and replace matrix multiplications by LTFHE-based shift-accumulation operations in TCN. As a result, SHE maintains the same inference accuracy but greatly reduces the inference latency to 9.3s. We noticed that SHE requires only 2.0K multiplicative depth (MD) which is much smaller than our LTFHE MD budget, so we can use a deeper network architecture (DSHE) with more convolutional and activation layers. DSHE obtains $99.77\%$ accuracy and spends 124.9s in an inference. We also reported the inference latency values of fully-TFHE-based TCN, SHE and DSHE. Compared to leveled-TFHE-based counterparts, they obtain much longer inference latency because of the computationally expensive bootstrapping operations.

The size of encrypted message that a client sends to cloud can be calculated by $MSize = PN \times SN \times PS$, where $PN$ is the pixel number of the input image; $SN$ means the polynomial number in a ciphertext; and $PS$ indicates the size of a polynomial. $PN$ is dependent on the dataset, while $SN$ and $PS$ are related to the cryptosystem parameters. For a MNIST image, $PN = 28 \times 28 = 784$. We quantized the inputs to 5-bit, so SHE encrypts each pixel in one MNIST image by 5 polynomials ($SN = 5$). In LTFHE, $PS = 32KB$. Totally, one MNIST image is encrypted into $784 \times 5 \times 32KB = 122.5MB$. Compared to other non-TFHE cryptosystems, SHE transfers much less message data between the client and the server. Typically, the encryption and decryption latency is proportional to the encrypted message size [4].

| Scheme | HOPs | $CC_{Add}$ | $PC_{Mult}$ | $CC_{Mult}$ | $PC_{Shift}$ | $CC_{Com}$ | HGOPs | Depth | MSize | EDL(s) | FIL(s) | LIL(s) | Acc(%) |
|---|---|---|---|---|---|---|---|---|---|---|---|---|---|
| GHE | 6.4M | 3.2M | 3.2M | 15K | 0 | 0 | - | - | - | 61.6 | - | 1628 | 62.1 |
| Lola | 137K | 61K | 61K | 15K | 0 | 0 | - | - | - | 5.7 | - | 730 | 74.1 |
| NED | 2.4G | 1.2G | 1.2G | 212K | 0 | 0 | 0 | - | 1.8GB | 21.8 | - | 127K† | 91.50 |
| FCN | 610M | 350M | 350M | 64K | 0 | 0 | 0 | 69.8K | 1.6GB | 8.8 | - | 39K† | 76.72 |
| TCN | 8.7M | 4.4M | 4.4M | 0 | 0 | 16K | 2.8G | 25.1K | 160MB | 0.0018 | 37M† | 31K† | 92.54 |
| **SHE** | 4.4M | 4.4M | 13K | 0 | 4.4M | 16K | 211M | 5.2K | 160MB | 0.0018 | 2.7M† | **2258** | **92.54** |
| **DSHE** | 68M | 68M | 98K | 0 | 68M | 131K | 3.3G | 13.7K | 160MB | 0.0018 | 42.5M† | **12041** | **94.62** |

Table 5: The CIFAR-10 results (Abbreviations are the same as Table 4; † denotes that we ran only its first 3 layers, while FIL and LIL values are made by projections).

## 5.2 CIFAR-10

Only NED, GHE, Lola and FCN can support CIFAR-10. As Table 5 shows, although GHE and Lola obtain shorter inference latency by shallow CNN architectures on CIFAR-10, compared to a full-precision unencrypted model, they degrade the inference accuracy by $> 20\%$, due to their PAA layers. Compared to NED with high degree polynomial approximation activations, FCN reduces the inference accuracy by $16.2\%$ but shortens the inference latency by $69.1\%$. With the same architecture, TCN reduces the activation computing overhead, but it requires longer convolution latency. Overall, it reduces the inference latency by $20.5\%$. However, the $ReLU$ activations and max poolings increase the inference accuracy to $92.54\%$. Compared to TCN, SHE reduces the number of $PC_{Mult}$ by $99.7\%$. As a result, it improves the inference latency by $92.7\%$ over TCN by performing only LTFHE shift-accumulation operations. Because SHE requires only 5.2K MD which is much smaller than our

LTFHE MD budget (32K), we can use a deeper network, DSHE, to increase the inference accuracy to 96.62% and the inference latency to 12041s.

| Network | Scheme | HOPs | $CC_{Add}$ | $PC_{Mult}$ | $PC_{Shift}$ | $CC_{Com}$ | HGOPs | Depth | MSize | EDL(s) | FIL(s) | LIL(s) | $Acc_I$(%) | $Acc_M$(%) |
|---|---|---|---|---|---|---|---|---|---|---|---|---|---|---|
| AlexNet | TCN | 0.3G | 0.14G | 0.14G | 0 | 0.66M | 38G | **42.3K** | 7.7GB | 0.07 | - | - | - | - |
| | SHE | 0.14G | 0.14G | 0.4M | 0.14G | 0.34M | 5.5G | 6.8K | 7.7GB | 0.07 | 72M† | 89K† | 54.17 | 63.24 |
| ResNet | TCN | 0.7G | 0.36G | 0.36G | 0 | 2.47M | 100G | **96.4K** | 7.7GB | 0.07 | - | - | - | - |
| | SHE | 0.36G | 0.36G | 1.1M | 0.36G | 0.49M | 15G | 13.7K | 7.7GB | 0.07 | 195M† | 0.23M† | 66.8 | 67.29 |
| Shuffle | TCN | 56M | 0.14G | 0.14G | 0 | 1.37M | 7.9G | 27.1K | 7.7GB | 0.07 | 102M† | 126K† | 69.4 | 71.32 |
| | SHE | 28M | 28M | 83K | 28M | 275K | 1.1G | 3.9K | 7.7GB | 0.07 | 14M† | **18K** | **69.4** | **71.32** |

Table 6: The ImageNet and Medical dataset results ($Acc_I$ means the inference accuracy of ImageNet, while $Acc_M$ denotes the inference accuracy of the medical dataset. The other abbreviations are the same as Table 4; † denotes that we ran only its first 3 layers, while FIL and LIL values are made by projections).

## 5.3 ImageNet and Medical Datasets

No prior LHECNN model can infer an entire ImageNet image, because of its prohibitive computing overhead. FCN [4] can compute only the last 3 layers of the model when inferring ImageNet. In Table 6, SHE uses the architectures of AlexNet, ResNet-18 and ShuffleNet for inferences on ImageNet. For AlexNet and ResNet-18, TCNs need >32K MD overhead which is larger than our LTFHE MD budget. Therefore, TCN cannot work on them. On the contrary, SHE needs 1 day and 2.5 days to test an ImageNet image by AlexNet and ResNet-18, respectively. Particularly, SHE requires only 5 hours to infer an ImageNet image and achieves 69.4% top-1 accuracy by the ShuffleNet topology. For the medical dataset, it obtains 71.32% inference accuracy. Besides shorter latency and higher accuracy, SHE enables a much deeper architecture under a fixed LTFHE MD budget, since the LTFHE shifts increase little MD.

| Scheme | HOPs | $CC_{Add}$ | $PC_{Mult}$ | $PC_{Shift}$ | $CC_{Com}$ | HGOPs | Depth | MSize | EDL(s) | FIL(s) | LIL(s) | PPW |
|---|---|---|---|---|---|---|---|---|---|---|---|---|
| TCN | 576K | 270K | 270K | 0 | 36K | 75.7M | **143K** | 9.4MB | 0.014 | - | - | - |
| SHE | 336K | 270K | 30.4K | 243K | 36K | 24.4M | 30K | 9.4MB | 0.014 | 318K† | **576** | **89.8** |

Table 7: The Penn Treebank results (Abbreviations are the same as Table 4; † denotes that we ran only its first 3 layers, while FIL values are made by projections).

## 5.4 Penn Treebank

⊛ : Matrix Multiplication   ⊙ : Element-wise Multiplication   ⊕ : Addition   ⋘ : Shift

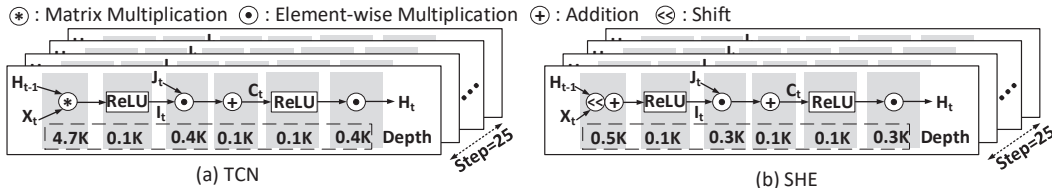

Figure 4: The MD overhead breakdown of the ReLU-based LSTM [15], and each LSTM layer repeats for 25 timesteps.

No prior LHECNN model supports LSTM, since it has a huge MD overhead. As Figure 4(a) shows, the MD overhead of the matrix multiplication in all time steps between $H_{t-1}$ and $X_t$ is $4.7K \times 25 = 117.5K$, which is already larger than the current 32K LTFHE MD budget. So TCN cannot successfully implement the LSTM architecture. As Figure 4(b) shows, We use SHE to build a LSTM network to predict the next word on Penn Treebank. SHE uses TFHE shifts and accumulations with small MD overhead to replace computationally expensive matrix multiplications, so it costs only 0.5K MD overhead to multiply $H_{t-1}$ with $X_t$. In Table 7, SHE costs totally 30K MD overhead, which is smaller than our LTFHE MD budget. The inference accuracy of LSTM on Penn Treebank is 89.8 Perplexity Per Word (PPW). Compared to the full-precision LSTM on plaintext data, it degrades the inference accuracy by only 2.1%. It takes 576s for SHE to conduct an inference on Penn Treebank.

## 6 Conclusion

In this paper, we propose SHE, a fast and accurate LTFHE-enable deep neural network, which consists of a $ReLU$ unit, a max pooling unit and a mixed bitwidth accumulator. Our experimental

results show SHE achieves the state-of-the-art inference accuracy and reduces the inference latency by $76.21\% \sim 94.23\%$ over prior LHECNNs on various datasets. SHE is the first LHE-enabled model that can support deep CNN architectures on ImageNet and LSTM architectures on Penn Treebank.

## Footnotes

*This work was supported in part by NSF CCF-1908992 and CCF-1909509.

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
