[Reviews · NeurIPS 2019]

Reviewer 1



Main contribution: The paper shows how to implement an accurate homomorphic ReLU and homomorphic max-pooling operation. This is achieved by combining the idea of logarithmic quantization followed by shifting and adding operations with the basic approach of TFHE (Fast Fully Momorphic Encryption over the Torus). [BTW the abbreviation for TFHE is never expanded and explained properly in the main paper]. Further they also note that 5 bit representations are sufficient for weights, but the intermediate results of accumulation need 16 bit representation to avoid degrading accuracy. Thus they propose mixed bitwidth accumulators to avoid unnecessary computational costs. By using these few key ideas the authors show how TFHE can now support fast matrix multiplications and convolutions which previously were extremely slow. Indeed the authors note that these kinds of operations were severely slowing down the execution of the previous state of the art method, FCN. Taken together their ideas dramatically speeded up computation while improving DNN accuracy, so this is a significant achievement. For example, previously, all the available methods including FCN used to take hundreds of seconds to process a single encryopted MNIST digit image through a very simplified 5 layer NN, and their accuracy was also very poor since they could not handle ReLUs or other such nonlinearities except through lousy and slow approximations. The improvements are so significant that the present paper/method is the first to be able to process ImageNet images or even run deep neural nets inference on homomorphically encrypted images. Further accuracy is also substantially better than previously available methods. In this sense the improvement over the current state of the art is very significant. Major disclaimer: This reviewer has not worked on cryptography and is not by any means an expert on privacy preserving deep neural nets. I would defer to experts to be able to validate that the improvement over the state of the art is truly this huge.

Reviewer 2



In my opinion the paper has minimal novelty in terms of techniques used and much of it is "obvious" to the homomorphic encryption community. The main contribution is using the TFHE scheme to get around the challenges of other arithmetic circuit based schemes. While this approach indeed works as the authors hope it to work, it comes with a massive cost in terms of message expansion. It seems unrealistic to use hundreds of megabytes, or even gigabytes, to store a single sample to classify. Due to the large size, this technique seems useless in an online setting. Overall I like the presentation of the paper. The performance analysis is quite detailed and compares to multiple prior works. I would think that these tables and benchmarks will be used in the future as reference points. However, I would like to point out that the comparisons are not exactly fair. Different papers and implementations use different security level, and some optimize for throughput rather than latency. It could be good to make this very clear in the paper.

Reviewer 3



strengths: -introduces fast new implementation for homomorphic encrypted neural networks -first paper to implement LSTMs with homomorphic encryption -good comparison with previous works. weaknesses: -no consideration for approximate number schemes in related work. -no support for float numbers. -At many points in the paper, it is not clear if unecrypted model is a model with PAA or a model with ReLU activation. -what is TCN? the abbreviation is explained way too late into the paper -Tables in chapter 5 are overloaded and abbreviations used are not explained properly. -Figure 3a does not highlight that the shift operation is cheap. - Although the authors claim they implement ImageNet for the first time, it is very slow and accuracy is very low; "SHE needs 1 day and 2.5 days to test an ImageNet picture by AlexNet and ResNet-18, respectively" and accuracy is around 70%

[Author Response · NeurIPS 2019]

We thank all reviewers for their careful reading of the manuscript and their constructive comments.

**Reviewer-1, Q1 readability & reproducibility**: We will elaborate all abbreviations, e.g., TFHE, in the next version of our draft. We will also release the attached code in the supplementary files into public repositories.

**Reviewer-7, Q1 using the logarithmic quantization and TFHE homomorphic encryption in neural networks to evaluate DNNs on encrypted input data is not exactly new**: Although the logarithmic quantization and TFHE homomorphic encryption are not proposed by us, combining them together to accelerate the inferences of Homomorphic-Encryption-enabled models is based on our new and key observation that the LTFHE shift operations are cheap. We also would like to emphasize that the other homomorphically encrypted shift operations, e.g. B/FV, FV-RNS and HEAAN shifts, are equivalent to homomorphic multiplications, and thus not cheap.

**Reviewer-7, Q2 practical usefulness when input data size is big**: First, all privacy-preserving deep learning models share the problem of the large communication overhead between client and server. Compared to Multiple Party Computation, Homomorphic Encryption has already significantly reduced the communication overhead between client and server. When using Multiple Party Computation, a prior work DeepSecure has to exchange 722GB data between client and server for only a 5-layer CNN inference on a tiny MNIST image. Second, our SHE uses TFHE to reduce the message size by $\sim 10\times$ over the state-of-the-art Homomorphic-Encryption-enabled models. We believe the 123MB input message size of SHE for a MNIST image and the 160MB input message size of SHE for a CIFAR-10 image are practical for privacy-preserving deep learning. Please notice that, during a Homomorphic Encryption inference, except the encrypted input message and the encrypted prediction result, there is no more communication between client and server.

**Reviewer-7, Q3 fair comparison for security level, latency and throughout**: Faster Cryptonets have the 128-bit security level, while Cryptonets and DiNN achieve the 80-bit security level. Our proposed SHE can obtain the 152-bit security level. Based on the white paper on Homomorphic Encryption Security Standardization (Martin Albrecht, et al, "Homomorphic Encryption Security Standard", HomomorphicEncryption.org, Toronto, Canada, 2018), a larger bit number indicates a higher security level. More details on the security level and configurations of SHE are described at the beginning of Section 4. We showed a detailed compassion on the latency values of various Homomorphic-Encryption-enabled models in Section 5. Based on the paper of Faster Cryptonets, in the setting of Machine Learning as a Service, it is not common for a user to submit 4096 images for homomorphically encrypted inferences. Therefore, we did not provide a detailed comparison on the throughput. But TFHE also supports the vertical and horizontal packing to batch 4096 input ciphertexts into a single cipertext, so that the processing throughput can be significantly boosted. We will add the throughput comparison in the next version of our draft.

**Reviewer-8, Q1 no consideration for approximate number schemes in related work**: We will add approximate number schemes, e.g. E2DM (Jiang, et al. "Secure Outsourced Matrix Computation And Application to Neural Networks." CCS 2018.), in the related work of the next version of the draft. E2DM uses the approximate-number technique to improve the latency and throughput of CryptoNets at the expense of obvious accuracy loss. In contrast, our SHE enables deeper neural networks on much larger encrypted input data with negligible accuracy loss. Based the E2DM paper and our draft, SHE is actually faster and more accurate than E2DM.

**Reviewer-8, Q2 no support for floating point numbers**: Because of the error tolerance, compared to the full-precision model, the fixed point quantization on neural networks can produce lossless accuracy. Fixed point quantized neural networks greatly reduce the message size and computing overhead during homomorphically encrypted inferences. Almost all the state-of-the-art homomorphic-encryption-enabled neural networks such as Cryptonets and faster Cryptonets focus on only fixed point quantized neural networks.

**Reviewer-8, Q3 unencrypted model with polynomial approximation activations or ReLU activations**: Our unencrypted model is trained with ReLU activations and performs inferences with ReLU activations.

**Reviewer-8, Q4 TCN and abbreviations in Section 5**: We will elaborate all abbreviations in the next version of our draft. TCN is defined in Section 5.1. We used the TFHE cryptosystem to implement the network architecture of faster Cryptonets by LTFHE-based ReLU activations, max poolings and matrix multiplications. We called this scheme TCN.

**Reviewer-8, Q5 Figure 3a does not highlight the shift operation is cheap**: We will highlight that a shift operation is cheap, i.e., each LTFHE shift only costs $\sim 100ns$ on a core of our CPU baseline.

**Reviewer-8, Q6 ImageNet is slow and inaccurate**: Because stacking polynomial approximation activation layers leads to a distortion on the output distribution of the following batch normalization layer, prior homomorphic-encryption-enabled models cannot be "deep" enough to work on ImageNet. Besides AlexNet and ResNet-18, we also built ShuffleNet for ImageNet in Section 5.3. One inference of SHE ShuffleNet takes 5 hours with 69.4% top-1 accuracy. At least, this is the very first try to deploy a homomorphic-encryption-enabled model to inferences on large ImageNet.

[Meta-Review · NeurIPS 2019]

This paper deals with the problem of adding privacy to the inference pipe-line of neural networks. Following previous results in this field, they propose the use of Homomorphic Encryption (HE). The authors use a different HE scheme then previous authors did which allows them to compute ReLUs and other activations that were only approximated in previous studies. They manage to do that while preserving relatively good computation time. This is a significant contribution as it suggest an alternative approach to the approaches used before. The authors may wish to include more recent results in Table 4. Some of this paper were published after the submission deadline but other were available even before this deadline. [1] Boemer and others, “nGraph-HE: A Graph Compiler for Deep Learning on Homomorphically Encrypted Data” [2] Boemer and others, “nGraph-HE2: A High-Throughput Framework for Neural Network Inference on Encrypted Data” [3] Brutzkus and others “Low Latency Privacy Preserving Inference”